# Active Teacher Selection for Reward Learning

**Rachel Freedman**                                            *rachel.freedman@berkeley.edu*
*Department of Computer Science and Electrical Engineering*
*University of California, Berkeley*

**Justin Svegliato**                                            *jsvegliato@berkeley.edu*
*Department of Computer Science and Electrical Engineering*
*University of California, Berkeley*

**Kyle Wray**                                            *k.wray@northeastern.edu*
*Khoury College of Computer Sciences*
*Northeastern University*

**Stuart Russell**                                            *russell@berkeley.edu*
*Department of Computer Science and Electrical Engineering*
*University of California, Berkeley*

**Reviewed on OpenReview:** *https://openreview.net/forum?id=9OEy68av40*

## Abstract

Reward learning techniques enable machine learning systems to learn objectives from human feedback. A core limitation of these systems is their assumption that all feedback comes from a single human teacher, despite gathering feedback from large and heterogeneous populations. We propose the *Hidden Utility Bandit* (HUB) framework to model differences in teacher rationality, expertise, and costliness, formalizing the problem of learning from multiple teachers. We develop a variety of solution algorithms and apply them to two real-world domains: paper recommendation systems and COVID-19 vaccine testing. We find that *Active Teacher Selection* (ATS) algorithms outperform baselines by actively selecting when and which teacher to query. Our key contributions are 1) the HUB framework: a novel mathematical framework for modeling the teacher selection problem, 2) ATS: an active-learning based algorithmic approach that demonstrates the utility of modeling teacher heterogeneity, and 3) proof-of-concept application of the HUB framework and ATS approaches to model and solve multiple real-world problems with complex trade-offs between reward learning and optimization.

## 1 Introduction

Specifying objective functions for machine learning systems is challenging, and misspecified objectives can be hacked (Pan et al., 2022; Skalse et al., 2022) or incentivise degenerate behavior (Zhuang & Hadfield-Menell, 2020; Thomas & Uminsky, 2020). *Reward learning* techniques such as reinforcement learning from human feedback (RLHF) enable state of the art ML systems to instead *learn* appropriate objectives by observing and interacting with human teachers (OpenAI, 2023; Anthropic, 2023; Touvron et al., 2023; Google, 2023). However, almost all deployed systems still rely on a *single-teacher assumption*: they assume feedback is generated by one canonical human teacher with fixed noise properties, even though in practice it is aggregated from a large, heterogeneous pool of humans. For example, Stiennon et al. (2020), Bai et al. (2022) and Ouyang et al. (2022) assume that all feedback comes from a single teacher, despite finding that annotators and researchers actually disagree 23% to 37% of the time.

This mismatch between formalism and reality is becoming increasingly problematic. In realistic settings, the humans that teach AI systems vary in expertise, attentiveness, and capabilities, annotation platforms mix

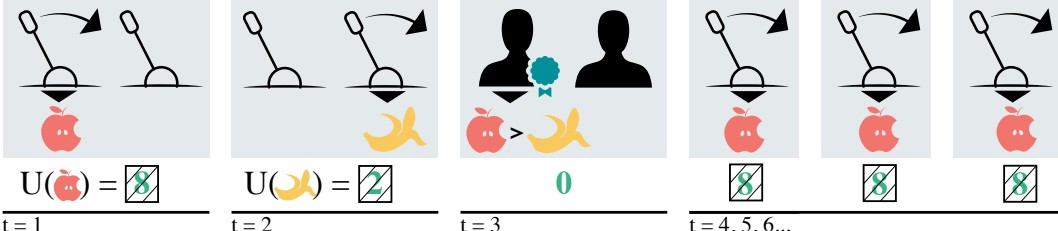

Figure 1: A simple *Hidden Utility Bandit (HUB)* with two arms and two teachers. The agent pulls the first arm, observes an apple, and receives the apple's utility of 8 without observing it. The agent then pulls the second arm, observes a banana, and receives the banana's utility of 2 without observing it. Because these utilities are hidden, the agent foregoes the opportunity for utility on the third timestep to ask the expert teacher which fruit is better. The expert replies that apples are better than bananas, so the agent pulls the first arm to maximize apples for all remaining timesteps.

crowdworkers with domain experts, and safety-critical systems must weigh the benefits of high-quality experts against their higher cost and scarcity. Prior work shows that reward learning is highly sensitive to incorrect assumptions about feedback generation (Hong et al., 2022; Freedman et al., 2021; Skalse & Abate, 2022; Milli & Dragan, 2020), and in particular that unrecognized heterogeneity in teacher rationality and context can significantly distort learned reward functions (Shirali et al., 2025; Daniels-Koch & Freedman, 2022). Nevertheless, current systems lack a framework for modeling differences between teachers and reasoning about *which* teacher to query for feedback *when* to query at all.

We formalize and study this *teacher selection problem*: given access to multiple teachers with different levels of reliability and cost, how should a reward-learning agent decide *which* teachers to query *when*? Good solutions must balance taking actions to gather utility according to the current reward model and querying teachers to improve that reward model, and determine when to query expensive, highly accurate teachers over cheaper, noisier ones. We address this gap by introducing the *Hidden Utility Bandit* (HUB) framework for reward learning from multiple teachers (Section 4). A HUB instance consists of a set of items with utility values that are hidden from the agent (but visible to teachers), a set of arms that stochastically produce items when pulled, and a set of teachers with a shared but unknown utility function. Figure 1 shows a sample solution to a simple HUB problem.

The HUB framework strictly generalizes standard multi-armed bandits by making arm utilities latent and learnable only through teacher feedback (Section 2), while remaining more structured and tractable than fully general cooperative inverse reinforcement learning formulations (Section 3). We derive upper and lower bounds on the number of teacher queries needed to recover the hidden utilities with high confidence, characterizing how query complexity scales with teacher noise, the number of items, and the required confidence level (Section 4.2). We deliberately assume that the utility function parameter space is small enough to compute precise Bayesian belief updates, even though in practice reward functions are often approximated by neural networks. This allows us to isolate the effect of teacher selection on inference and downstream performance without conflating it with optimization or function-approximation issues.

Building on this formalism, we propose *Active Teacher Selection* (ATS), a class of solution methods that convert the HUB to a partially observable Markov decision process (POMDP) and make teacher selection part of the planning problem (Section 5). ATS maintains a belief over utilities and arm outcome distributions and uses a Monte Carlo tree-search planner (POMCPOW) with tailored rollout policies to select actions that approximately maximize expected discounted utility. Unlike naive two-phase procedures that first gather feedback and then exploit a fixed estimate, ATS interleaves exploration and exploitation, naturally trades off cost against accuracy across teachers, and does not require problem-specific exploration schedules or hand-picked "reference" teachers. We also show how to estimate teacher rationality parameters from empirical preference data, so that ATS can operate even when teacher noise levels are not known a priori (Section 6.2).

Since there are no existing solutions to the novel HUB problem, we introduce multiple families of baseline methods and evaluate these against ATS on a realistic recommendation task in Section 5.2[1]. ATS outperforms methods with fixed exploration windows, demonstrating the usefulness of selecting *when* to query teachers, and ATS with specific teacher selection outperforms general teacher selection, demonstrating the usefulness of selecting *which* teacher to query. As a proof-of-concept, we also demonstrate application of this framework to the real-world problem of evaluating COVID-19 vaccines with expensive and unreliable tests in Section 6. Our HUB framework and ATS algorithm demonstrate the importance of leveraging differences between teachers to learn accurate reward models and will facilitate future work on scalable reward learning algorithms that learn accurate, robust and value-aligned models from diverse teachers.

## 2    Preliminaries

**Reward Learning**    The goal of *reward learning* is to estimate a function $\hat{\mathcal{U}} : \mathcal{I} \to \mathbb{U}$ mapping alternative items from the set $\mathcal{I}$ to scalar utility or reward values in settings where they cannot be observed directly. We focus on the most common type of reward learning, *preference learning*, in which the reward function is inferred from human preference comparisons between pairs of alternatives. For example, the teacher in Figure 1 compares the alternatives {*apple*, *banana*} and selects *apple*. The AI can then infer that *apple* likely has higher reward.

Because human teachers can make mistakes, they are typically modeled as noisily rational, choosing alternative $i$ over $j$ with probability $\Pr(i \succ j)$ rather than deterministically. In the reward learning literature, human feedback is most often modeled as Boltzmann-rational (Rajkumar & Agarwal, 2014; Jeon et al., 2020), where the probability that a teacher with rationality parameter $\beta \in [0, \infty)$ prefers item $i$ to $j$ is:

$$\Pr(i \succ j; \beta, \mathcal{U}) = \frac{\exp(\beta \mathcal{U}(i))}{\exp(\beta \mathcal{U}(i)) + \exp(\beta \mathcal{U}(j))}, \tag{1}$$

where $\mathcal{U} : \mathcal{I} \to \mathbb{R}$ gives the true utility of all items in set $\mathcal{I}$. While Boltzmann-rationality does not fully model all nuances of human decision-making (Lindner & El-Assady, 2022), it does capture the important property that humans are more likely to make mistakes when $|\mathcal{U}(i) - \mathcal{U}(j)|$ is small and therefore the comparison is harder (Barnett et al., 2023), which likely accounts for its empirical success and pervasive use in practice. It is now the dominant model for safety finetuning large language models (see for example (Bai et al., 2022)), so we will use it in this work.

While existing reward inference work typically assumes that the teacher rationality hyperparameter $\beta$ is known, we find this to be an unrealistic assumption, and therefore show how to infer an estimate $\hat{\beta}$ in Section 6.2. Reward learning systems also typically assume that all feedback is generated by a single Boltzmann-rational teacher model as described in Equation 1, despite differences in teacher expertise (Daniels-Koch & Freedman, 2022) or context (Pitis et al., 2024; Siththaranjan et al., 2023). In this work we relax this assumption, modeling differences between teachers and the process of selecting between them.

**Teacher Selection**    Given a set of different teachers, the reward inference agent must choose which teacher(s) to query so as to accurately infer the underlying reward function $\mathcal{U}$. We develop a framework to formalize this problem in Section 4, building off of the related frameworks of multi-armed bandits and partially-observable markov decision processes.

*Multi-armed bandits* (MAB) are stateless sequential decision-making problems (Robbins, 1952) that model choices between fixed sets of alternatives. At each timestep the agent chooses one of $K$ arms, each with a distribution over utilities. When the agent pulls arm $k \in K$, it receives utility sampled from arm $k$'s distribution $u \sim \mathcal{D}^k$. The agent's goal is to maximize its expected cumulative utility. Our framework is similar, though arm utilities are hidden (as in many real-life applications), and the agent must learn about them from teacher preferences (as in reward learning).

*Partially observable Markov decision processes (POMDPs)* are sequential decision-making problems where elements of the world state (in this case, the underlying reward function $\mathcal{U}$), can be hidden from the

---

[1]Code to reproduce our experiments is available at `github.com/[redacted]/ATS`.

agent (Littman et al., 1995). A POMDP problem is a tuple $\langle \mathcal{S}, \mathcal{A}, \mathcal{T}, \mathcal{R}, \mathcal{O}, \Omega, \gamma \rangle$, where $\mathcal{S}$ and $\mathcal{A}$ are the state and action spaces, $\mathcal{T}$ and $\mathcal{R}$ are the transition and reward functions, and $\gamma$ is the discount factor. At time $t$, the agent begins in state $s_t$, takes action $a_t$, transitions to state $s_{t+1}$ determined by $\mathcal{T}(s_t, a_t)$ and receives reward $r_t = R(s_t, a_t, s_{t+1})$. Rather than observing states directly, the agent observes an observation $\omega_{t+1}$ from the observation space $\mathcal{O}$ determined by the observation function $\Omega(s_{t+1}, a_t)$. A POMDP solution is a policy that balances inferring the underlying state and acting in the environment to maximise expected cumulative reward.

While calculating this solution is typically intractable, approximate POMDP algorithms can perform well. Partially observable Monte Carlo planning (POMCP) algorithms produce time-efficient online solvers that form a belief tree of fixed depth then use rollouts to estimate the values of the leaf nodes (Silver & Veness, 2010). In this work we will show how to formulate the teacher selection problem as a POMDP, then solve it using partially observable Monte Carlo planning with observation widening (POMCPOW), a POMCP-style algorithm that uses a weighted particle filter to efficiently produce approximate solutions for problems with large state spaces (Sunberg & Kochenderfer, 2018). We develop problem-specific rollout policies to customize POMCPOW in Appendix 9.

## 3 Related Work and Contributions

To the best of our knowledge, the teacher selection problem is novel and so there are no pre-existing solutions. However, related work studies learning from heterogeneous teachers, aggregating differing teacher values, and human-AI cooperation on assistance problems. We overview research in each of these areas, discuss their relationship to our formalism and algorithms, and then outline the novel contributions of this work.

### 3.1 Related Work

**Teacher Heterogeneity**  Several prior papers model teacher heterogeneity in reward inference, but few of these models permit active teacher selection. Siththaranjan et al. (2023) model teacher populations as distributions rather than collections of individuals, which prevents selecting individual teachers. Pitis et al. (2024) finetune reward models to consider information about context that may cause variation in annotator preferences and Poddar et al. (2024) learn individual latent variables for each user, but these assume that the context or user is fixed. Shirali et al. (2025) prove that it is impossible to represent a mixture of Boltzmann-rationality models (representing a set of diverse population of teachers) with a single Boltzmann model, underscoring the importance of modeling teacher heterogeneity, but assume that the teacher distribution is already chosen.

The HUB framework differs from this prior work along three axes. First, it models teachers as individuals with distinct rationality and cost parameters, rather than as draws from a distribution as in (Siththaranjan et al., 2023), enabling active selection of which teacher to query. Second, it treats teacher selection as a sequential decision-making problem with cost-aware planning, rather than assuming the teachers are given as in (Pitis et al., 2024), (Poddar et al., 2024), and (Shirali et al., 2025). Third, it interleaves inference and action as an online sequential process, rather than performing reward learning in a separate phase as in (Siththaranjan et al., 2023), and (Pitis et al., 2024), (Poddar et al., 2024).

The most closely related works are (Daniels-Koch & Freedman, 2022) and (Barnett et al., 2023), both of which model teacher selection amongst teachers with variation in expertise. However, Daniels-Koch & Freedman (2022) use a simple heuristic for teacher selection (maximizing rationality) rather than weighing expected informativeness against cost as the active teacher selection (ATS) algorithm we propose does. Section 5 discusses the limitations of this heuristic. Barnett et al. (2023) develop a greedy, value-of-information-based teacher selection algorithm that selects the teacher whose single query would most reduce expected belief error. This myopic strategy can be understood as a single-step-lookahead approximation to ATS's multi-step planning. However, it operates in a pure-inference setting without arm-pulling, query costs, or an exploitation objective. ATS extends this approach to multi-step planning in the HUB domain, which requires jointly deciding when to gather information (explore) and when to collect reward (exploit).

**Value Heterogeneity** In this work, we address the case where teachers share an underlying utility function, and differences in their preference feedback arises from differences in their expertise or context on the problem. However, in some cases teacher *values* themselves vary. Fleisig et al. (2023) and Zhang et al. (2024) model value disagreement amongst human annotators, Siththaranjan et al. (2023) and Shirali et al. (2025) explore the shortcomings of aggregating these diverse perspectives implicitly, and Freedman (2026) propose a concrete method for explicitly representing and integrating diverse stakeholder values when filtering AI output. Deciding how to aggregate these diverse sets of values is an open problem in social choice theory, and thus beyond the scope of this work, but see Conitzer et al. (2024) for discussion of how to integrate social choice theory and reward learning.

**Assistance Games** *Assistance games* are problems in which an AI system and a human must collaborate to achieve the human's goals (Hadfield-Menell et al., 2016; Malik et al., 2018). The AI system cannot observe the human's goals directly, so optimal human behavior often involves teaching the AI Milli & Dragan (2020). HUB problems can be viewed as a specific class of assistance games in which there are multiple teachers, but they can only act (by providing feedback) when the agent requests it (by querying them). However, assistance games are DEC-POMDPS, which are NEXP-complete and thus functionally intractable (Bernstein et al., 2002). By fixing the teacher policy and arm distributions, the HUB framework reduces the problem to a much more tractable POMDP with a stationary transition function. Optimal AI solutions to the assistance game balance inference and control to produce qualitatively valuable behaviors, such as only asking the human questions when necessary (Shah et al., 2020). Our ATS algorithm leverages this insight by actively deciding when to query teachers, rather than doing so on a fixed schedule as in traditional reward learning.

## 3.2 Contributions

We state our contributions and how we overcome limitations in prior work:

1. We explicitly model differences in teacher rationality and query cost, rather than assuming that differences in teacher feedback are due to additional "context" of arbitrary form. This model allows algorithms to reason about tradeoffs between cost and informativeness of querying specific teachers.

2. We define the *hidden utility bandit* (HUB), a novel problem formalism for the teacher selection problem (see Section 4). The HUB is more expressive than a MAB, but can be converted to a POMDP for tractability.

3. We define a procedure for estimating the teacher rationality parameter $\beta$, rather than requiring it to already be known (see Section 6.2).

4. We apply active learning to develop a novel class of *active teacher selection* (ATS) solution methods that leverage teacher models to efficiently trade off the informativeness of feedback and query costs (see Section 5).

5. We provide a case study demonstrating how to apply the HUB framework and ATS algorithm to a real-world vaccine testing problem (see Section 6.

## 4 Hidden Utility Bandits

We design the *Hidden Utility Bandit* (HUB) framework to formalize the problem of reward learning from multiple teachers. Formally, a HUB is a partially-observable sequential decision-making problem consisting of a set of items (each with a distinct utility), a set of arms (each with a fixed distribution over items), and a set of Boltzmann-rational teachers (each with a rationality parameter and cost). At each step of the HUB problem, the agent chooses between either pulling an arm, observing an item sampled from that arm's distribution and receiving *but not observing* that item's utility, or querying a teacher, receiving feedback modulated by that teacher's rationality parameter but incurring that teacher's query cost.

| Paper Category | Relevance |
|---|---|
| Application | 8 |
| Benchmark | 5 |
| Theory | 1 |

| Professor | Rationality | Cost |
|---|---|---|
| Professor 1 | 0 | 0 |
| Professor 2 | 0.01 | 0 |
| Professor 3 | 50 | 0 |

| Conference | ICLR | | | ICML | | | AAAI | | |
|---|---|---|---|---|---|---|---|---|---|
| | A | B | T | A | B | T | A | B | T |
| Paper Distribution | 0.8 | 0 | 0.2 | 0.2 | 0.8 | 0 | 0.2 | 0.2 | 0.6 |
| Expected Relevance | | 6.6 | | | 5.6 | | | 3.2 | |

Figure 2: Paper recommendation as a HUB problem. Paper categories (Application, Benchmark, Theory) are items ($\mathcal{I}$), professors are teachers with rationality ($\beta$) and cost ($F$) parameters, conferences are arms with distributions ($\mathcal{D}$), and relevance scores are utilities ($\mathcal{U}$). The goal is to recommend the most relevant conferences to read papers from.

**Definition 4.1.** A *hidden-utility bandit* **(HUB)** is a tuple $\langle \mathcal{I}, \mathcal{U}, \mathcal{C}, \beta, F, Q, \gamma \rangle$:

- $\mathcal{I}$ is a set of $N$ *items*, each with a hidden utility.
- $\mathcal{U} : \mathcal{I} \to [u_{\min}, u_{\max}]$ is a *utility function* over $\mathcal{I}$, where $\mathbb{U}$ is the utility function space.
- $\mathcal{C} = \{c^1, c^2, \ldots, c^K\}$ is a set of $K$ *arm choices*, each associated with an *arm distribution* $\mathcal{D}^k : \mathcal{I} \to [0,1]$ giving the probability of returning each item in $\mathcal{I}$, where $\mathbb{D} = \mathbb{D}^1 \times \mathbb{D}^2 \times \cdots \times \mathbb{D}^K$ is the joint arm distribution space over all arm choices $\mathcal{C}$.
- $\beta = \{\beta^1, \beta^2, \ldots, \beta^M\}$ is $M$ *teacher rationality parameters*.
- $F = \{f^1, f^2, \ldots, f^M\}$ is $M$ *teacher query costs*.
- $Q : \mathcal{I} \times \mathcal{I} \to [0,1]$ is a *query profile* that gives probabilities of picking queries in $\binom{\mathcal{I}}{2}$.
- $\gamma$ is a discount factor.

The agent can observe $\mathcal{I}, \mathcal{C}, \beta, F, Q$, and $\gamma$ but cannot observe the utility function $\mathcal{U}$ or the arm distributions $\mathcal{D}$. At each timestep $t$, the agent can select an arm choice $c_t \in \mathcal{C}$ or a teacher rationality parameter $\beta_t \in \beta$. If the agent pulls an arm choice $c_t \in \mathcal{C}$, it observes an item $i_t$ sampled from the arm distribution $\mathcal{D}^{c_t}$ and receives but does *not* observe the utility $u_t = \mathcal{U}(i_t)$. Conversely, if the agent queries a teacher with rationality parameter $\beta_t \in \beta$, it receives and observes an item pair $(i, j)$ sampled from the query profile $Q$, a preference $p_t$ sampled from Bernoulli($P$) given the probability $P = \Pr(i \succ j; \beta_t, \mathcal{U})$ in Equation 1, and the teacher query cost $u_t = f^{\beta_t}$. In this work we use a simple query profile that selects queries randomly in order to focus on the teacher selection problem. However, the query profile can easily be adjusted to incorporate more complex sampling strategies.

The agent's objective is to maximize the expected discounted sum of utilities $\mathbb{E}[\Sigma_{t=0}^{\infty} \gamma^t u_t]$. To do this, it must balance querying teachers to learn about the utility function with selecting bandit arms to earn utility. Standard reward learning systems alternate between fitting a reward model to teacher feedback and learning a policy using the reward model on a predefined schedule. However, the HUB framework allows the agent to interweave these processes to optimize performance.

**Example: Paper Recommendation**

For example, consider a recommender system tasked with recommending AI conference papers to a PhD student. Imagine that there are three types of papers (Application, Benchmark, Theory) and three conferences (ICLR, ICML, AAAI) that produce different distributions of paper types. Some types of papers are more relevant to the student's research than others, but the recommender system doesn't initially know how to distinguish these. Instead, it must learn which paper types are most relevant by asking professors, whose judgements vary from completely random ($\beta^1 = 0$) to highly accurate ($\beta^3 = 50$). Each day, the system either recommends a conference to the student, in which case a paper is sampled from that conference's distribution, and the system earns a hidden utility score representing that paper's type's relevance, or picks a professor to ask for feedback, in which case that professor compares two paper types and provides a preference. (For simplicity, we assume that the paper types the professor compares are chosen randomly, and that the only cost of asking a professor for feedback is the opportunity cost of the student not reading a paper that day.)

Applying the HUB framework, paper categories are the item set $\mathcal{I} = \{A, B, T\}$, relevance scores are the hidden utility function $\mathcal{U}$, conferences are arm choices $\mathcal{C} = \{c^1 = \text{ICLR}, c^2 = \text{ICML}, c^3 = \text{AAAI}\}$, and

---

**Algorithm 1** NAIVEHUBINFERENCE($\cdot$)

---

**Require:** HUB $\langle \mathcal{I}, \mathcal{U}, \mathcal{C}, \beta, F, Q, \gamma \rangle$, $u_{\min}$, $u_{\max}$, $T$ samples, $\beta^m$ of selected teacher
**Initialize:** frequency[$c$], frequency[$c$][$i$], frequency[$b$][$q$], preferences[$b$][$q$]

1: **for** $t = 1, \ldots, T$ **do**
2:     **if** sampleUniformly($\{$TRUE, FALSE$\}$) **then**
3:         sample $c \sim \mathcal{C}$                                           $\triangleright$ Sample arm uniformly at random
4:         sample $i \sim \mathcal{D}^c$                             $\triangleright$ Sample item from (unobserved) arm distribution
5:         frequency[$c$] $\leftarrow$ frequency[$c$] $+ 1$
6:         frequency[$c$][$i$] $\leftarrow$ frequency[$c$][$i$] $+ 1$
7:     **else**
8:         sample $b \sim \beta$                                      $\triangleright$ Sample teacher uniformly at random
9:         sample $q = (i, j) \sim Q$                       $\triangleright$ Sample query from query profile
10:       sample $p \sim$ Bernoulli($\Pr(i \succ j; b, \mathcal{U})$)     $\triangleright$ Sample preference given Equation 1
11:       frequency[$b$][$q$] $\leftarrow$ frequency[$b$][$q$] $+ 1$
12:       preferences[$b$][$q$] $\leftarrow$ preferences[$b$][$q$] $+ p$
13: $\hat{D}^c(i) \leftarrow \frac{\text{frequency}[c][i]}{\text{frequency}[c]}$   $\forall c \in \mathcal{C}, i \in \mathcal{I}$                $\triangleright$ Estimate arm distributions
14: $\hat{P}(b, q) \leftarrow \frac{\text{preferences}[b][q]}{\text{frequency}[b][q]}$   $\forall b \in \beta, q \in Q$          $\triangleright$ Estimate preference probabilities
15: $\Delta_{ij} = -\frac{1}{\beta^m} \ln \left[ \frac{1}{\hat{P}(\beta^m, q = (i,j))} - 1 \right]$   $\forall i, j \in \mathcal{I}$
16: $(x, y) \leftarrow \arg\max_{x,y} \left[ \Delta_{xy} \right]$                             $\triangleright$ Find indices of maximum element
17: $\hat{\mathcal{U}}(y) \leftarrow u_{\min}$,   $\hat{\mathcal{U}}(i) \leftarrow \left[ \frac{u_{\max}}{u_{\max} - u_{\min}} \right] \Delta_{iy} + u_{\min}$   $\forall i \in \mathcal{I} \setminus \{y\}$     $\triangleright$ Estimate utilities

---

professors are teachers with rationality $\beta = \{\beta^1 = 0, \beta^2 = 0.01, \beta^3 = 50\}$. Figure 2 shows an example paper recommendation HUB problem.[2] In this problem, the system must learn how much more relevant *Application* papers are than *Benchmark* papers in order to determine whether to recommend *ICLR* or *ICML*. Without this information, the system cannot distinguish between cases where $\mathcal{U}(A) = 8$ (indicating that the expected relevance of *ICLR* is greater than *ICML*) and where $\mathcal{U}(A) = 6$ (indicating the reverse). Interestingly, in this example it will sometimes be more informative for the recommendation system to query the noisy Professor 2 over the more rational Professor 3, because the frequency with which a noisy teacher prefers a lower-reward item over a higher-reward one gives information about the *magnitude* of the difference between the rewards.

## 4.1 Naive HUB Inference

We first propose a simple baseline algorithm: naive HUB inference (Algorithm 1). Naive HUB inference explores randomly for a fixed number of timesteps $T$ (lines 1-12), performs a frequentist estimate of the expected utility of each arm based on those $T$ observations (lines 13-17), then greedily selects the highest-expected-utility arm for the remainder of the task. For example, a paper recommendation system following the naive algorithm will randomly recommend conferences and choose professors for the first $T$ days, pause and calculate which conference's papers appear to be most relevant according to the professor feedback it has gotten so far, and then recommend only the conference that it infers to be most useful going forward.

This allows the agent to infer hidden information: the joint arm distribution $\mathcal{D}^{\mathcal{C}} = (\mathcal{D}^1, \mathcal{D}^2, \ldots, \mathcal{D}^K)$ (common to stochastic multi-armed bandit problems) and utility function $\mathcal{U}$ (unique to the HUB). Morever, despite the simplicity of Algorithm 1, it is possible to prove that it converges to the ground truth utility function $\mathcal{U}^*$ and arm distribution set $\mathcal{D}^{\mathcal{C}*}$ in the limit of infinite queries. We prove the following theorem in Appendix 8.1:

**Theorem 4.2.** *If the predicted utility function $\hat{\mathcal{U}}$ and the predicted arm distribution $\hat{\mathcal{D}}^{\mathcal{C}}$ are estimated by executing Algorithm 1 with $T$ samples, then $\hat{\mathcal{U}} \to \mathcal{U}^*$ and $\hat{\mathcal{D}}^{\mathcal{C}} \to \mathcal{D}^{\mathcal{C}*}$ as $T \to \infty$.*

However, exploring randomly for a fixed number of timesteps and querying a fixed teacher may be suboptimal. By maintaining and updating an internal belief over the hidden information, the agent can instead query teachers only when teacher feedback is necessary to update its belief.

---

[2]Example relevance scores and paper category compositions were selected arbitrarily.

## 4.2 Query Sample Complexity

We evaluate the efficiency of learning hidden utilities $\mathcal{U}$ from teacher queries by computing upper and lower bounds on teacher query sample complexity. Theorem 4.3 (proved in Appendix 8.2) upper bounds queries required for a confident estimation as a function of the number of correct classifications required to learn $\mathcal{U}$, which is nontrivial to calculate but can be estimated from our naive baseline experiments (described in Section 6.2). Theorem 4.4 (proved in Appendix 8.3) lower bounds queries required to receive correct classifications for each query pair at least once, assuming that the teacher is queried about each pair until a correct answer is received.

**Theorem 4.3.** *We can bound the maximum number of teacher queries $t$ required to learn the hidden utilities $\mathcal{U}$ with $\overline{\kappa}$ confidence in the worst case as $t \leq \frac{r(1-p)}{p\overline{\kappa}}$, where $r$ is the number of successful classifications from teachers required to learn the hidden utilities, and $p = \min_{\beta, \Delta_{ij}} \frac{1}{1+\exp(-\beta(\Delta_{ij}))}$ is the worst-case teacher accuracy.*

**Theorem 4.4.** *We can bound the minimum number of teacher queries $t$ required to receive a correct answer for each query pair with $\overline{\kappa}$ confidence in the worst case as $t \geq \frac{\log(\frac{k}{p})N(N-1)}{2\log(1-p)}$, where $N$ is the number of items, and $p = \min_{\beta, \Delta_{ij}} \frac{1}{1+\exp(-\beta(\Delta_{ij}))}$ is the worst-case teacher accuracy.*

Both bounds depend on the worst-case teacher accuracy $p$, which is a function of the product of the teacher's rationality $\beta$ and the utility gap $\Delta_{ij}$. In order for these bounds to be informative, $p$ must be $> 0.5$. This holds under two assumptions: (1) at least one teacher has rationality $\beta > 0$, which is already implicit in any setting where querying teachers is useful, since a fully random teacher ($\beta = 0$) provides no information; and (2) the query profile $Q$ includes at least one item pair $(i, j)$ with $\mathcal{U}(i) \neq \mathcal{U}(j)$, ensuring a positive utility gap $\Delta_{ij}$. In practice, the query profile can be designed to avoid comparing items with near-identical utilities.

# 5 Active Teacher Selection

The *Active Teacher Selection (ATS)* algorithm solves the HUB problem efficiently by maintaining a belief over the utility function and arm distributions, and choosing when to query teachers. This allows it to only query when required for decision-relevant belief updates. This alleviates the need to set the problem-specific hyperparameters in Algorithm 1 for exploration ($T$) and teacher selection ($\beta^m$).

ATS also actively selects *which* teacher to query. This is useful because some teachers are "noisy" ($\beta < \infty$), and therefore their preference probability $\Pr(i \succ j; \beta, \mathcal{U})$ correlates with the difference in utility between $i$ and $j$. This means that, for HUB problems such as the paper recommendation problem in Figure 2, it is sometimes more informative for ATS to select teachers with *lower* $\beta$ values (Barnett et al., 2023).

## 5.1 ATS Algorithm

The ATS algorithm has two phases: first convert the HUB to a simplified partially observable Markov decision process (POMDP) (Littman et al., 1995), then solve it using a Monte Carlo POMDP solver with HUB-specific custom rollout policies.

**Phase 1: Constructing the HUB-POMDP** The HUB-POMDP state contains the HUB utility function and arm distributions. The HUB-POMDP reward function gives the *expected* utility of each arm according to this state. The full HUB-POMDP is specified in Definition 5.1.

**Definition 5.1.** A **hidden utility bandit POMDP (HUB-POMDP)** is a tuple $\langle \mathcal{S}, \mathcal{A}, \mathcal{T}, \mathcal{R}, \Omega, \mathcal{O} \rangle$:

- $\mathcal{S} = \mathbb{U} \times \mathbb{D}$ is the state space: the state $s \in \mathcal{S}$ is a tuple $\langle \mathcal{U}, \mathcal{D}^{\mathcal{C}} \rangle$ that is fixed.
- $\mathcal{A} = \mathcal{C} \cup \beta$ is the action space: the arm choices $\mathcal{C}$ and teachers $\beta$.
- $\mathcal{T} : \mathcal{S} \times \mathcal{A} \to \mathcal{S}$ is the stationary transition function: $\mathcal{T}(s, a) = s \ \forall_{s \in \mathcal{S}} \ \forall_{a \in \mathcal{A}}$.
- $\mathcal{R} : \mathcal{S} \times \mathcal{A} \to \mathbb{R}$ is the reward function:

$$\mathcal{R}(s, a) = \begin{cases} \Sigma_{i \in \mathcal{I}} \mathcal{U}(i) \mathcal{D}^a(i) & \text{if } a \in \mathcal{C} \\ -f^a & \text{if } a \in \beta \end{cases}$$

- $\Omega : \mathcal{I} \cup \mathbb{P}$ is the observation space: the items $\mathcal{I}$ and query-preferences $\mathbb{P} = \mathcal{I} \times \mathcal{I} \times \{0, 1\}$.
- $\mathcal{O} : \mathcal{A} \times \Omega \to [0, 1]$ is the observation function:

$$\mathcal{O}(a, \omega) = \begin{cases} \mathcal{D}^a(i) & \text{if } a \in \mathcal{C} \\ Q(i, j) \Pr(i \succ j; \beta^m = a, \mathcal{U}) & \text{if } a \in \beta \end{cases}$$

Teacher selection can be *general* or *specific*. Under specific selection, the agent chooses which teacher to query. The HUB-POMDP's action space contains all $M$ teachers, $\mathcal{A} = \mathcal{C} \cup \beta$, as shown in the HUB-POMDP above. Under general selection, the agent chooses *when* to query a teacher, but as in RLHF cannot choose *which* teacher to query. The HUB-POMDP's action space is modified to contain a single general teacher selection action, $\mathcal{A} = \mathcal{C} \cup \{\beta^g\}$.

These alternatives offer a tradeoff: general selection reduces the state space size and computational complexity while specific selection provides the agent with additional control over its feedback. Our experimental results (reported in Section 5.3) indicate that specific greatly outperforms general teacher selection, so we will use ATS with specific teacher selection unless otherwise specified.

**Phase 2: Solving the POMDP**  While exact POMDP solutions are typically intractable, approximate POMDP algorithms often perform well. *Partially observable Monte Carlo planning (POMCP)* algorithms produce time-efficient online solvers that form a belief tree of fixed depth and use rollouts to estimate leaf node values (Silver & Veness, 2010). *POMCP with observation widening (POMCPOW)* uses a weighted particle filter to efficiently produce approximate solutions for problems with large state spaces (Sunberg & Kochenderfer, 2018), so we adapt it to the HUB-POMDP with specialized rollout policies. We describe and compare candidate rollout policies that we designed specifically for the HUB problem in Appendix 9. ATS with the custom *best arm* rollout policy performs best, so we use that POMCPOW variant.

**Optimality**  In a standard multi-armed bandit, regret is defined relative to an oracle that always pulls the best arm. However, this definition is insufficient for the HUB problem because it does not account for the value and cost of teacher queries. The natural comparator is the Bayes-optimal POMDP policy, which itself queries teachers strategically. ATS inherits asymptotic convergence guarantees from POMCPOW, meaning that as the number of simulations per decision step increases, the selected action does converge to the Bayes-optimal action for the current belief state (Silver & Veness, 2010; Sunberg & Kochenderfer, 2018). ATS therefore approximates the optimal teacher selection and arm-pulling policy for any HUB instance, given sufficient computation per step.

However, characterizing *finite-time* approximation bounds specific to the HUB-POMDP is non-trivial. It requires jointly reasoning about the cost of information and the value of future actions, and couples the exploration and exploitation problems in a way that classical bandit regret decompositions do not handle. We therefore validate ATS's finite-time performance empirically. Our experiments in Section 5.2 demonstrate that it outperforms baselines with practical computation budgets.

### 5.2 Empirical Performance

We evaluate active, naive and random algorithms on the paper recommendation HUB task described in Section 4. We find that, while both active and naive algorithms successfully identify the most relevant conference in expectation (Figure 3b), ATS with specific teacher selection best balances querying teachers with recommending papers, achieving the highest average discounted cumulative reward (Figure 3a), and most accurately learning relevance scores (Figure 4). In further experiments, we compare rollout simulation policies (Appendix 9), and examine the impact of varying teacher query costs (Appendix 10).

**Algorithms**  We fix ATS to use the *best arm* rollout policy and *specific* teacher selection for this set of experiments. (We later compare specific and general teacher selection in Section 5.3.) To our knowledge, the HUB problem is novel and has no solutions in prior literature, so we construct multiple families of baseline methods (*random* and *naive*) for comparison. *Random* algorithms select actions uniformly at random from a given set. We evaluate a random algorithm that selects actions from the entire action space, as well as one that selects only arms.

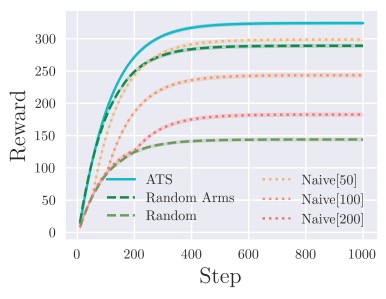
(a) Discounted cumulative reward

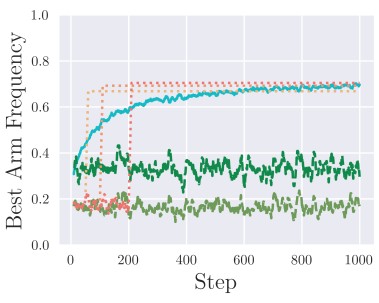
(b) Frequency of pulling best arm

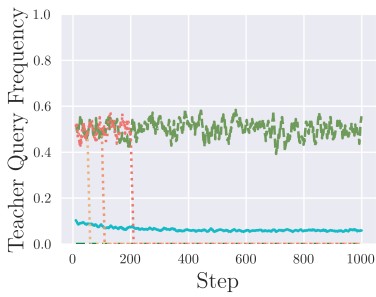
(c) Frequency of querying teacher

Figure 3: Comparison of ATS, naive and random algorithms. ATS best maximizes discounted reward (a) and identifies the highest-reward arm more often than most baselines and comparably with Naive[100] and Naive[200], which explore more and earn less reward (b). ATS initially queries teachers less often than naive baselines, but continues querying teachers throughout the episode (c). All data is averaged across 25 runs on 20 HUB problems and smoothed over 10 steps.

*Naive* algorithms choose randomly amongst pulling arms and querying the selected teacher for $T$ timesteps, use these observations to estimate the arm distributions and utility function (using Algorithm 1), then pull the arm with the highest estimated expected utility at each timestep. This baseline is chosen to represent the dominant approach in current RLHF practice: collect a fixed batch of human preference data, fit a reward model offline, then deploy a policy trained on that model. Although prior work has advocated for online iterative RLHF (Dong et al., 2024), most deployed systems still follow this pre-collected-data paradigm (Kaufmann et al., 2024; Casper et al., 2023). The performance gap between ATS and naive baselines demonstrates the benefit of replacing this rigid schedule with adaptive interleaving of inference and action.

**Experiments** We evaluate all algorithms for 25 runs of 1000 steps on 20 paper recommendation tasks. Each task is a HUB with $\mathcal{I}$, $\mathcal{C}$, and $\beta$ as described in Section 4 and a unique tuple $\langle \mathcal{U}, \mathcal{D}^\mathcal{C} \rangle$. $\mathbb{U}$ and $\mathbb{D}$ are discretized, and each task's $\langle \mathcal{U}, \mathcal{D}^\mathcal{C} \rangle$ is chosen such that $c^1$ has the highest expected relevance ($\mathbb{E}[\mathcal{U}(i \sim c^1)] > \mathbb{E}[\mathcal{U}(i \sim c^2)] \geq \mathbb{E}[\mathcal{U}(i \sim c^3)]$) and all paper distributions are different and non-deterministic ($\mathcal{D}^j \neq \mathcal{D}^k \; \forall_{j,k \in \mathcal{C}}$ and $\mathcal{D}^c(i) \neq 1.0 \; \forall_{i \in \mathcal{I}, c \in \mathcal{C}}$). Naive algorithms require problem-specific hyperparameters $\beta^m$ and $T$, so for these experiments we select the intermediate of 3 teachers ($\beta^m = \beta^2$) and test a range of exploration horizons ($T \in [50, 100, 200]$).

**Results** While all non-random algorithms successfully identify the most relevant conference in expectation (Figure 3b), ATS also learns the most accurate relevance scores (Figure 4), and uses this inference to earn the highest average discounted cumulative reward on average (Figure 3a).

Figure 3b shows how often each algorithm learns to pull the best HUB arm and therefore recommend the most relevant conference over the course of training. All HUB solution methods (ATS, Naive[50], Naive[100], Naive[200]) successfully identify the most relevant conference, recommending it about three times as often as they would if they were behaving randomly ("Random" baseline, light green line) and about twice as often as if they were blindly recommending conferences ("Random Arms" baseline, dark green line). This indicates that the HUB formalism can be used to accurately represent the paper recommendation problem.

While all solution methods identify the best arm, ATS does so most efficiently, querying teachers sparingly even at the start of the task (Figure 3c) and best optimizing the HUB objective of expected discounted cumulative reward (Figure 3a). Moreover, ATS forms the most accurate estimates of the utility function and expected conference relevance scores (Figure 4) after 1000 timesteps, while continuing to explore and potentially improve this estimate by occasionally querying teachers and recommending other conferences (Figure 5a). In contrast, Naive algorithms stop learning after their hand-specified exploration horizon (Figure 5b), and Random algorithms never learn at all (Figure 5c). This demonstrates the benefits of actively selecting *when* to query teachers, as in ATS, rather than following a predefined RLHF schedule.

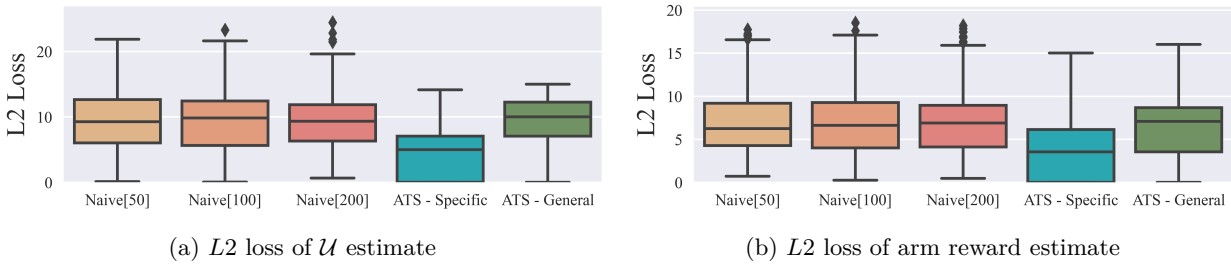

(a) $L2$ loss of $\mathcal{U}$ estimate

(b) $L2$ loss of arm reward estimate

Figure 4: Accuracy of reward learning using ATS (with specific and general teacher selection) and naive algorithms (with exploration parameters of 50, 100, and 200). ATS with specific teacher selection learns both the underlying utility function (a) and the expected rewards of each arm (b) much more accurately than ATS with general teacher selection and naive algorithms. The middle line is the median, boxes are the IQR, whiskers are 1.5 times the IQR, and diamonds are outliers.

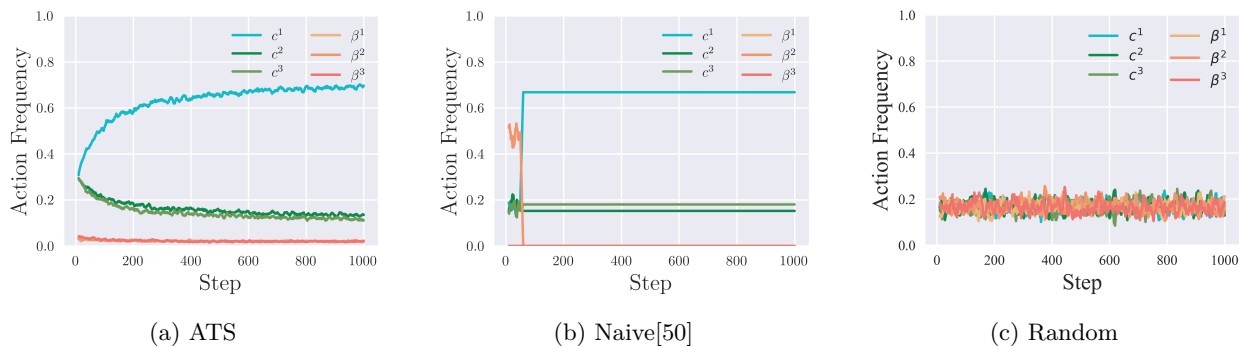

(a) ATS

(b) Naive[50]

(c) Random

Figure 5: Mean action frequencies for various algorithms. $c$ actions are arm pulls and $\beta$ actions are teacher queries. Data is averaged across 25 runs of 20 HUB problems and smoothed over 10 steps.

### 5.3 Specific v. General selection

We further investigate the impact of actively selecting *which* teacher to query ("specific" selection) over having the teacher chosen randomly ("general" selection). Standard RLHF systems do not allow the agent to select which teacher to query and are therefore most akin to general selection. However, we find that the additional control afforded by specific selection allows ATS to make more informative queries, and consequently to earn higher reward.

Figure 6 compares the performance of the ATS algorithm with *specific* teacher selection (in which ATS can choose which teacher to query) against *general* teacher selection (in which the teacher is chosen randomly) on the conference recommendation task. Figure 6a shows that ATS with specific teacher selection consistently earns higher expected reward than ATS with general teacher selection across the course of training. Figure 6b shows that ATS with general teacher selection queries all arms roughly equally, typically failing to identify the one with highest expected reward. The performance gap between specific and general selection demonstrates the importance of modeling differences between teachers and actively selecting *which* teacher to query.

## 6 Case Study: Vaccine Testing

Bandit-type problems are commonly used to model medical treatment investigation, so as a proof-of-concept we apply the HUB framework to a real-world medical problem: evaluating vaccines for COVID-19. In this case study, the "teachers" are COVID-19 tests rather than human experts, hilighting the flexibility of the HUB model. Some types of tests are more expensive to run than others, so we calculate testing costs.

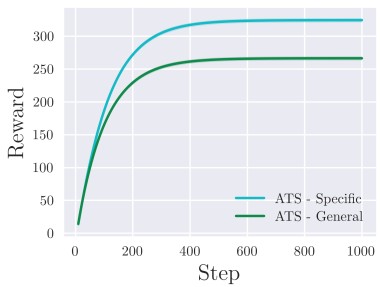 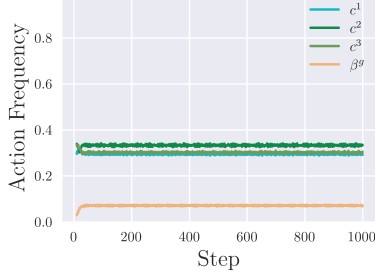

(a) Specific teacher selection outper-     (b) ATS with general teacher selec-
forms general teacher selection.        tion doesn't identify the best arm.

Figure 6: Performance of ATS with specific and general teacher selection. All data is averaged across 25 runs on 20 HUB problems, smoothed over 10 steps, and discounted with $\gamma = 0.99$.

We assume that the vaccine trial has two goals: 1) identify the most effective vaccine, and 2) reduce the rate of COVID-19 infection over the course of the trial. We find that active teacher selection is the only algorithm to accomplish both goals. The Random Arms baseline algorithm, which vaccinates patients ("pulls arms") at every timestep and never runs tests ("queries teachers"), actually earns slightly higher reward on average (likely because it avoids high testing costs). However, because it never queries a teacher, it never learns about the underlying utility function and consequently fails to identify the most effective vaccine.

## 6.1 HUB experiments

COVID-19 vaccine testing is complicated by the difficulty of evaluating whether a patient is infected: many infections are asymptomatic, and other illnesses cause similar symptoms. There are many ways to test whether patients have COVID-19, including symptom surveys, antigen tests, and RT-PCR tests, but the choice of which to use is nontrivial, since these alternative tests vary widely in accuracy and cost.

The HUB framework directly models these challenges. Let the item set be easily observable patient symptoms, $\mathcal{I} = \{\text{None}, \text{Cough}, \text{Fever}\}$. The "arms" are vaccine candidates, $\mathcal{C} = \{c^1 = \text{VaccineA}, c^2 = \text{VaccineB}, c^3 = \text{NoVaccine}\}$, and the "teachers" are COVID-19 test types, $\{\text{Survey}, \text{Antigen}, \text{RT-PCR}\}$. Surveys are the least accurate but least expensive, while RT-PCR tests are the most accurate and most expensive.

**Experiments** We fix the parameters to the values reported in Figure 7, which are derived from real-world data and estimates. Specifically, we estimate the US dollar cost of surveys at \$1.20 (accounting for 10 minutes of time at the US federal minimum wage of \$7.25), antigen tests at \$42, and RT-PCR tests at \$62 (median prices reported by (Lo et al., 2023)), then scale these costs by 0.05. We construct arm distributions where patients display the most frequent and severe symptoms with no vaccination, and the least symptoms with Vaccine A, and a utility function where symptoms that have a greater chance of indicating COVID-19 infection have lower scores. Finally, we discuss and demonstrate how to infer the teacher noise parameters $\beta$ in Section 6.2. We evaluate all algorithms for 25 runs of 1000 steps on this COVID-19 task. $\mathbb{U}$ and $\mathbb{D}$ are more finely discretized than in the recommendation HUB to permit more realistic values, so the resulting HUB-POMDP has 5 times more states and is more challenging to solve.

**Results** Figure 8 summarises the performance of various HUB algorithms on the COVID-19 vaccine testing task. We find that both ATS and the Random Arms algorithm earn high reward during training (Figure 8a), but of these only ATS successfully identifies the most effective vaccine ($c^1$ in Figure 8b). Random Arms earns high reward during training because it vaccinates at every timestep (and never pays for a test), but fails to learn which vaccine is most effective, which is a core goal of the vaccine trial. In contrast, the Naive baselines identify the best vaccine, but conduct excessive costly tests during training. ATS strikes the best balance, identifying the best vaccine while conducting a minimum of costly tests.

| Symptoms | Utility |
|----------|---------|
| None | 8.0 |
| Cough | 3.0 |
| Fever | 0.5 |

| Test | Rationality | Cost |
|------|-------------|------|
| Survey | 0.36 | -0.006 |
| Antigen | 1.32 | -0.21 |
| RT-PCR | 2.54 | -0.31 |

| Vaccine | Vaccine A | | | Vaccine B | | | No Vaccine | | |
|---------|---|---|---|---|---|---|---|---|---|
| | N | C | F | N | C | F | N | C | F |
| Symptom Distribution | 0.9 | 0.1 | 0 | 0.6 | 0.3 | 0.1 | 0.5 | 0.3 | 0.2 |
| Expected Utility | 7.5 | | | 5.75 | | | 5.0 | | |

Figure 7: COVID-19 vaccine testing as a HUB problem. Symptoms (None, Cough, Fever) are items ($\mathcal{I}$), tests are teachers with rationality ($\beta$) and cost ($F$) parameters, and vaccines are arms ($\mathcal{C}$) with the specified distributions over patient symptoms ($\mathcal{D}$).

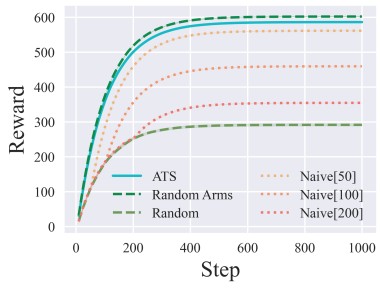

(a) Discounted cumulative reward

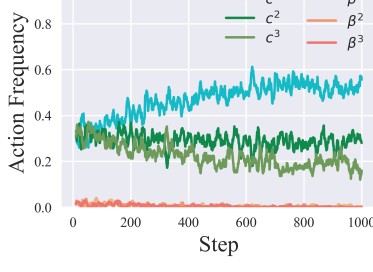

(b) ATS action frequencies

Figure 8: Performance of all algorithms and ATS action frequencies on the COVID-19 vaccine testing problem. Random Arms and ATS both earn high reward from frequently vaccinating participants (a), though only ATS additionally identifies the most effective vaccine (b).

## 6.2 Teacher Noise Inference in ATS

RLHF systems typically assume that the teacher rationality parameters $\beta$ are known. However, as this is sometimes unrealistic, we show in Theorem 6.1 that $\beta$ can also be estimated from preference data. Specifically, given $\Pr(i \succ j; \beta_m, \mathcal{U})$, it is possible to estimate $\hat{\beta}_m = \frac{1}{z}\beta_m$, where $z$ is a scaling factor determined by $\mathcal{U}$. $z$ is based on the difference $\Delta_{ij} = \mathcal{U}(i) - \mathcal{U}(j)$, so as long as the same comparison pair $(i, j)$ is used, all teacher rationality estimates will be on the same scale. (They can be calculated directly if $\Delta_{ij}$ happens to be known for a specific $(i, j)$.)[3] We prove the theorem below in Appendix 8.4.

**Theorem 6.1.** *Given two items $i, j \in \mathcal{I}$ where $\mathcal{U}(i) < \mathcal{U}(j)$ and the preference probability $P = \Pr(i \succ j; \beta_m, \mathcal{U})$ from Equation 1 we can estimate $\hat{\beta}_m$ as $\hat{\beta}_m = \ln\left(\frac{1}{P} - 1\right)$. If $\Delta_{ij}$ is known, we can further calculate $\beta_m = z \cdot \hat{\beta}_m$, where $z = -\Delta_{ij}^{-1}$.*

We evaluate this procedure in simulation by setting $\beta = \{0.01, 1.0\}$, running a random policy for 1000 timesteps, estimating $\{\hat{\beta}_1, \hat{\beta}_2\}$, and scaling the estimate so that the greatest value is equal to 1.0. We observe a mean squared error of only 0.061 across 100 simulations, indicating that this procedure is accurate.

For the vaccine testing case study, we can estimate $\beta$ by gathering real-world data on the sensitivity of COVID-19 symptom surveys (Rufino et al., 2023), antigen tests (Harmon et al., 2021), and RT-PCR tests (Binny et al., 2023), interpret this sensitivity as the probability $P$ of the test "preferring" a patient with no COVID-19 ($\mathcal{U} = u_{max}$) to a patient with definite COVID-19 ($\mathcal{U} = u_{min}$), and let $\Delta_{ij} = u_{min} - u_{max}$. Then we calculate $\beta_m$ for each test type using Theorem 6.1. These values are reported in Figure 7 and used in our experiments.

---

[3]Note that it is also possible to directly add $\beta$ to the state space of the HUB-POMDP and then solve it, but this increases the size of the state space and makes the problem less tractable.

## 7 Conclusion

We formalized the teacher selection problem in reward learning and proposed a solution method that actively selects teachers based on their query cost and expertise. Our empirical results underscore the applicability of this framework to real-world problems, as well as the importance of modeling human teachers as distinct entities and actively choosing both *when* and *which* to query.

**Scalability**  Our ATS algorithm uses Bayesian belief updates over discretized utility and distribution spaces, which limits scalability to larger problems. ATS incorporates POMCPOW to solve the HUB-POMDP, which improves scalability by using particle filtering to maintain approximate belief representations and using progressive widening to limit the branching factor. Future work could further improve POMDP solver scalability using variational inference and state abstraction.

In RLHF settings where the reward function is parameterized by a neural network rather than a discrete utility table, extending the HUB framework would require replacing belief updates with approximate posterior inference over network parameters (for example, using Laplace approximation or ensembles). While this is a significant engineering challenge, the conceptual framework of modeling teacher heterogeneity and planning over teacher queries to balance informativeness against cost transfers directly. Moreover, by making explicit the cost-accuracy tradeoffs between teachers, our HUB framework offers a principled way to reason about teacher selection that can inform simpler heuristic strategies at any scale.

**Societal Impact**  While this work is intended to improve AI alignment by learning accurate, robust and value-aligned reward models from diverse teachers, the explicit teacher modeling could be vulnerable to bias. The HUB framework assigns rationality parameters $\beta$ to teachers, which in practice could correlate with demographic factors such as seniority, education, or cultural background. If rationality estimates are biased, the system may systematically under-query certain groups of teachers, potentially excluding important perspectives. Similarly, the cost structure could encode or reinforce existing inequities if, for example, expert teachers are more expensive because they belong to privileged groups. Practical mitigation strategies include fairness constraints on query allocation and auditing rationality estimates for demographic bias.

**Limitations and Future Work**  The purpose of this work is to investigate the novel problem of selecting teachers in reward learning, so we empirically evaluate tasks where *learning* the utility function is more challenging than *optimizing* it. However, many real-world tasks, such as language model finetuning, also involve challenging optimizations across enormous state spaces. Future work should combine our methods and insights with existing work on state abstractions and hierarchical options to scale the HUB formalism and ATS method to more complex domains.

## Acknowledgments

This work was supported by a grant from Open Philanthropy to the Center for Human-Compatible Artificial Intelligence at UC Berkeley. Rachel Freedman is supported by a Cooperative AI Foundation (CAIF) PhD Fellowship.

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

# 8 Proofs

## 8.1 State Estimation (Theorem 4.2)

*Theorem* 4.2. If the predicted utility function $\hat{\mathcal{U}}$ and the predicted arm distribution $\hat{\mathcal{D}}^{\mathcal{C}}$ are estimated by executing Algorithm 1 with $T$ samples, then $\hat{\mathcal{U}} \to \mathcal{U}^*$ and $\hat{\mathcal{D}}^{\mathcal{C}} \to \mathcal{D}^{\mathcal{C}*}$ as $T \to \infty$.

*Proof (Sketch).* Since the number of arms is finite and they are pulled uniformly as $T \to \infty$, the number of times that a given arm $c^k$ is pulled approaches infinity. Since each pull samples an item from the true distribution $\mathcal{D}^{k*}$ i.i.d., the empirical distribution $\hat{\mathcal{D}}^k$ will approach $\mathcal{D}^{k*}$ in the limit of infinite pulls. This argument applies for all arms $c^k \in \mathcal{C}$, so $\hat{\mathcal{D}}^{\mathcal{C}} \to \mathcal{D}^{\mathcal{C}*}$ as $T \to \infty$. Similarly, in the limit of infinite queries, $\hat{P}(\beta, (i, j))$ will approach $P^*(\beta, (i, j)) = \Pr(i \succ j; \beta, \mathcal{U}^*)$, the true probability that teacher $b$ prefers item $i$ over item $j$, as determined by Equation 1. Given $\beta$, $(i, j)$ and $\hat{P}(\beta, (i, j))$ from the first $T$ timesteps, we can calculate $\Delta_{ij} = \hat{\mathcal{U}}(i) - \hat{\mathcal{U}}(j)$ using Equation 2 below.

$$\Pr(i \succ j; \beta^m, \mathcal{U}) = \frac{1}{1 + \exp(-\beta^m \Delta_{ij})} \tag{2}$$

$$\implies \Delta_{ij} = -\frac{1}{\beta^m} \ln\left[\frac{1}{\Pr(i \succ j; \beta^m, \mathcal{U})} - 1\right]. \tag{3}$$

Given $\Delta = [\Delta_{01}, \Delta_{02}, \dots, \Delta_{NN}]$, $u_{max}$ and $u_{min}$, we can calculate $\hat{\mathcal{U}}$ as described in Algorithm 1. $\hat{\mathcal{U}} \to \mathcal{U}^*$ as $\hat{P} \to P^*$, which occurs as $T \to \infty$. $\qquad\square$

## 8.2 Query Complexity Upper Bound (Theorem 4.3)

*Theorem* 4.3. We can bound the maximum number of teacher queries $t$ required to learn the hidden utilities $\mathcal{U}$ with $\overline{\kappa}$ confidence in the worst case as $t \leq \frac{r(1-p)}{p\overline{\kappa}}$, where $r$ is the number of successful classifications from teachers required to learn the hidden utilities, and $p = \min_{\beta, \Delta_{ij}} \frac{1}{1 + \exp(-\beta(\Delta_{ij}))}$ is the worst-case teacher accuracy.

*Proof.* Let $X \sim \text{NegBinom}(r, p)$.

Let $\overline{\kappa}$ represent the upper bound desired confidence.

We bound the maximum number of queries $t$ required to observe $r$ correct classifications with confidence $\overline{\kappa}$ as:

$$\Pr(X = t; r, p) \geq \overline{\kappa}$$

$$\frac{E[X]}{t} \geq \Pr(X = t; r, p) \geq \overline{\kappa} \qquad \text{applying the Markov Inequality}$$

$$\left( \frac{r(1 - p)}{p} \right) \cdot \left( \frac{1}{t} \right) \geq \overline{\kappa} \qquad \text{derived from the definition of the Negative Binomial Probability Mass Function}$$

$$\frac{r(1 - p)}{p\overline{\kappa}} \geq t$$

The worst case teacher accuracy is

$$p = \min_{\beta, \Delta_{ij}} \frac{\exp(\beta \mathcal{U}(i))}{\exp(\beta \mathcal{U}(i)) + \exp(\beta \mathcal{U}(j))} \qquad \text{from Equation 1}$$

$$= \min_{\beta, \Delta_{ij}} \frac{1}{1 + \exp(-\beta \Delta_{ij})}.$$

$\square$

## 8.3 Query Complexity Lower Bound (Theorem 4.4)

*Theorem* 4.4. We can bound the minimum number of teacher queries $t$ required to receive a correct answer for each query pair with $\overline{\kappa}$ confidence in the worst case as $t \geq \frac{\log(\frac{k}{p})N(N-1)}{2\log(1-p)}$, where $N$ is the number of items, and $p = \min_{\beta, \Delta_{ij}} \frac{1}{1 + \exp(-\beta(\Delta_{ij}))}$ is the worst-case teacher accuracy.

*Proof.* We bound the minimum number of queries $t$ required to observe the correct answer $i \succ j$ with confidence $\kappa$ as:

$$\Pr(X \geq t; p) \geq \kappa$$

$$(1 - p)^t p \geq \kappa$$

$$t \geq \frac{\log(\frac{\kappa}{p})}{\log(1 - p)}$$

Since there are $N$ items, there are $\frac{N(N-1)}{2}$ unique pairs. The minimum number of queries $t$ required to observe the correct answer for each pair with confidence $\kappa$ is therefore

$$t \geq \frac{\log(\frac{k}{p})N(N - 1)}{2\log(1 - p)}.$$

The worst case teacher accuracy is

$$p = \min_{\beta, \Delta_{ij}} \frac{\exp(\beta \mathcal{U}(i))}{\exp(\beta \mathcal{U}(i)) + \exp(\beta \mathcal{U}(j))} \qquad \text{from Equation 1}$$

$$= \min_{\beta, \Delta_{ij}} \frac{1}{1 + \exp(-\beta \Delta_{ij})}.$$

$\square$

### 8.4 $\beta$ Estimation (Theorem 6.1)

*Theorem 6.1.* Given two items $i, j \in \mathcal{I}$ where $\mathcal{U}(i) < \mathcal{U}(j)$ and the preference probability $P = \Pr(i \succ j; \beta_m, \mathcal{U})$ from Equation 1 we can estimate $\hat{\beta}_m = \frac{1}{z}\beta_m$ as in Equation 4. If $\Delta_{ij}$ is known, we can further calculate $\beta_m = z \cdot \hat{\beta}_m$, where $z = -\Delta_{ij}^{-1}$.

$$\hat{\beta}_m = \ln\left(\frac{1}{P} - 1\right). \tag{4}$$

*Proof (Sketch).* First, we define an affine mapping function $f_{a,b}(x) = ax + b$ such that $f_{a,b}(\mathcal{U}(i)) = 0$ and $f_{a,b}(\mathcal{U}(j)) = 1$. Lemma 8.1 shows that this is always possible when $\mathcal{U}(i) \neq \mathcal{U}(j)$ and furthermore that $a = \frac{-1}{i-j}$. Let $z$, $y$ be the parameters that make this mapping for these particular values of $\mathcal{U}(i)$ and $\mathcal{U}(j)$. Note that $z = \frac{-1}{i-j} = -\Delta_{ij}^{-1}$.

Next, suppose we have that $\beta'_m = \frac{1}{a}\beta_m$, it follows that:

$$
\begin{aligned}
P &= \Pr(i^0 \succ i^1; \beta_m, \mathcal{U}) \\
&= \frac{\exp(\beta_m \mathcal{U}(i))}{\exp(\beta_m \mathcal{U}(i)) + \exp(\beta_m \mathcal{U}(j))} && \text{(by Equation 1)} \\
&= \frac{\exp(\frac{\beta_m}{a} \cdot a\mathcal{U}(i) + \frac{\beta_m}{a}b)}{\exp(\frac{\beta_m}{a} \cdot a\mathcal{U}(i) + \frac{\beta_m}{a}b) + \exp(\frac{\beta_m}{a} \cdot a\mathcal{U}(j) + \frac{\beta_m}{a}b)} \\
&= \frac{\exp(\beta'_m \cdot (a\mathcal{U}(i) + b))}{\exp(\beta'_m \cdot (a\mathcal{U}(i) + b)) + \exp(\beta'_m \cdot (a\mathcal{U}(j) + b))} && \text{(by definition of } \beta'_m) \\
&= \frac{\exp(\beta'_m \cdot f_{a,b}(\mathcal{U}(i)))}{\exp(\beta'_m \cdot f_{a,b}(\mathcal{U}(i))) + \exp(\beta'_m \cdot f_{a,b}(\mathcal{U}(j)))} && \text{(by definition of } f_{a,b}) \\
&= \frac{\exp(0)}{\exp(0) + \exp(\beta'_m)} = \frac{1}{1 + \exp(\beta'_m)}.
\end{aligned}
$$

Finally, solving for $\beta'_m$ yields $\beta'_m = \frac{1}{z}\beta_m = \ln(\frac{1}{P} - 1) \quad \rightarrow \quad \beta_m = z \cdot \ln(\frac{1}{P} - 1)$. $\square$

**Lemma 8.1.** *Given any two numbers $m, n \in \mathbb{R}$ such that $m \neq n$, there exists an affine transformation $f_{a,b} : \mathbb{R} \to \mathbb{R}$ that maps the greater number to $1$ and the lesser number to $0$.*

*Proof (Sketch).* Suppose that $m > n$ without loss of generality. We therefore must solve the following system of equations: $f_{a,b}(m) = am + b = 1$ and $f_{a,b}(n) = an + b = 0$. The solution is $a = \frac{-1}{n-m}$ and $b = \frac{m}{n-m} + 1$, which always exists when $m \neq n$. $\square$

## 9 POMCPOW Rollout Policies

ATS solves the HUB-POMDP using *partially observable Monte-Carlo planning with observation widening* (POMCPOW) augmented with a custom rollout policy for estimating the value of leaf nodes in the search tree. We evaluate a *random action* rollout policy, which takes actions uniformly at random from $\mathcal{A} = \mathcal{C} \cup \beta$, a *random arm* rollout policy, which chooses arms uniformly at random from $\mathcal{C}$, and a *best arm* policy, which calculates which arm has the highest expected utility *according to the current belief $b$*, then always chooses that arm.

Since a utility-maximizing agent will choose arms more often if it believes them to have higher utility, the *best arm* policy rollouts most closely resemble the actions the actual policy would take from belief $b$, yielding the most accurate value estimates. As a result, ATS with best arm rollouts outperforms the alternatives on the paper recommender domain, as shown in Figure 9. Results are averaged across 25 runs on 20 different paper recommendation tasks.

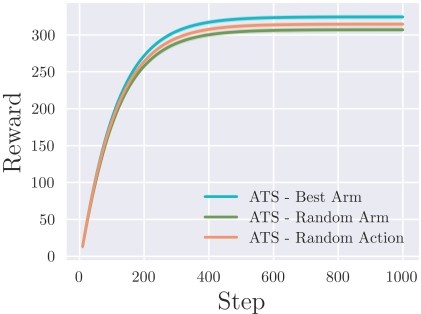

Figure 9: Performance of ATS with various rollout policies. The best arm rollout policy outperforms the random arm and random action rollout policies. All data is averaged across 25 runs on each of 20 HUB problems, smoothed over 10 steps, and discounted with $\gamma = 0.99$.

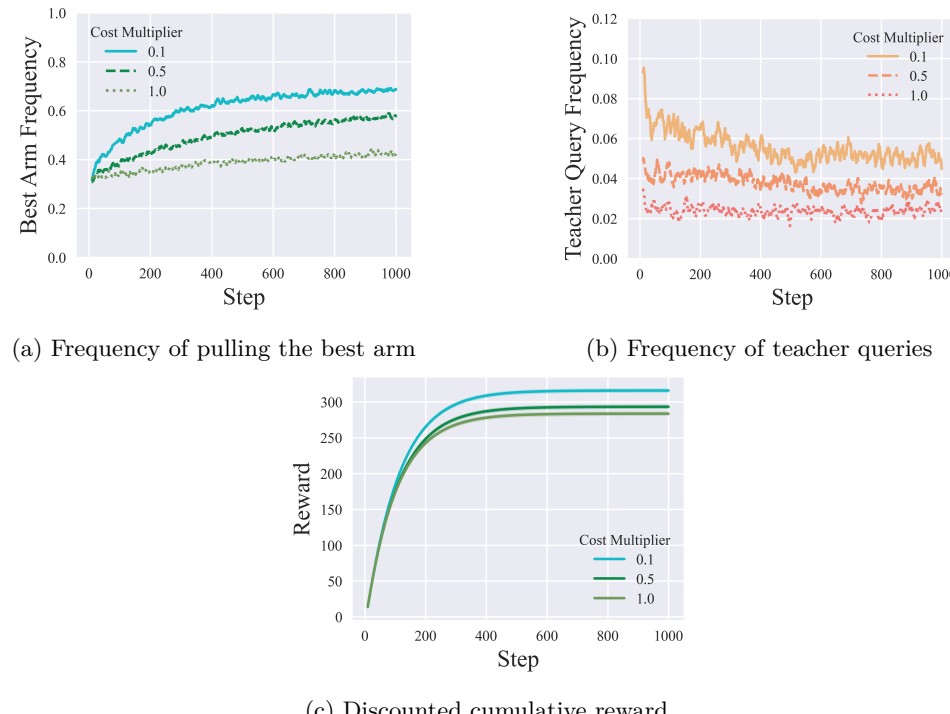

Figure 10: ATS behavior and performance varies with teacher query costs. Data is averaged across 25 runs on 20 paper recommendation HUB problems and smoothed over 10 steps.

## 10    HUB Cost effects

We investigate the impacts of teacher query cost on ATS performance by varying professor feedback costs in the paper recommendation domain. We set linear costs $F = \{-1, -2, -3\}$ and scale them by a *cost multiplier*. As in the other paper recommendation experiments, results are averaged across 25 runs on 20 different paper recommendation tasks.

We find that ATS responds rationally to changes in costs, querying teachers more sparingly (Figure 10b) and consequently identifying the best arm more slowly (Figure 10a as overall costs increase. This leads to a slight decrease in overall performance (Figure 10c).

