# OpenReview forum: "Active Teacher Selection for Reward Learning"
_TMLR — Accepted by TMLR_

### Review · Reviewer_qXrq · 2026-02-06

**Summary Of Contributions:**

**Summary**

This paper studies the problem of learning reward functions from multiple heterogeneous teachers and proposes the Hidden Utility Bandit (HUB) framework to formalize teacher selection under query costs. The authors model teacher heterogeneity via different rationality and cost parameters, and formulate the resulting decision problem as a POMDP, which is solved using an Active Teacher Selection (ATS) approach based on Monte Carlo planning. The paper provides theoretical analysis characterizing consistency and query complexity, and demonstrates the framework on paper recommendation and vaccine testing domains. Overall, the paper is clearly written, technically sound, and presents a formulation of a problem that has received limited explicit treatment in prior reward learning work.

**Strengths**

1. The paper identifies a realistic gap between standard single-teacher reward learning models and practical settings involving heterogeneous feedback.
2. The paper is easy to follow despite addressing multiple conceptual layers (reward learning, bandits, POMDPs) sources

**Weaknesses**

1. Relationship to prior work could be sharpened. While the paper discusses relevant literature on learning from heterogeneous teachers and active reward learning, the distinction between the proposed teacher selection problem and prior work could be articulated more explicitly to better position the novelty of the HUB formulation
2. Empirical evaluation is limited in scale and complexity. The experiments are confined to small, stylized domains with discrete state spaces and handcrafted parameters. While suitable as proof-of-concept demonstrations, they leave open questions about how the proposed approach would scale to more realistic reward learning settings.

**Audience:**

Yes

**Audience Explanation:**

The paper addresses a conceptual gap in reward learning and RLHF-style systems by explicitly formalizing the teacher selection problem under heterogeneous feedback and query costs. Researchers interested in reward learning, human-in-the-loop learning, and decision-making under uncertainty are likely to find the proposed framework and analysis informative, even if the current instantiation is limited to small-scale domains.

**Claims And Evidence:**

Yes

**Claims Explanation:**

The authors provide a formal framework and supporting analysis that align with their claims, and the empirical results (although toy-scale) consistently illustrate the qualitative advantages of active teacher selection over baseline approaches in the considered domains. While the theoretical analysis is not the primary focus of my expertise, the arguments appear internally consistent and are used appropriately to motivate and contextualize the proposed framework.

**Requested Changes:**

1. Clarify the distinction from related work. It would be helpful for the authors to more explicitly contrast the HUB framework with prior work on active reward learning from multiple teachers, particularly clarifying how teacher selection as a sequential decision-making problem differs from prior query selection or reward inference formulations.

2. Discuss scalability and practical extensions. While the current experiments are intentionally small-scale, a brief discussion (possibly in the limitations or future work section) outlining how the proposed ideas might extend to larger reward spaces, function approximation, or approximate belief updates would strengthen the paper.

---

> ### Author Response · Authors · 2026-02-22
> **Response to qXrq (part 1)**
>
> We sincerely thank the reviewer for their thoughtful and encouraging review. We address each requested change below, and will upload a revised paper after receiving all reviews.
>
> ---
>
> ## Change 1: Contrast related work
>
> We appreciate this suggestion and agree that making the novelty of the HUB framework more explicit will strengthen the paper. We will revise the "Teacher Heterogeneity" discussion in Section 3.1 to explicitly contrast the HUB framework with prior work on reward learning from multiple teachers.
>
> The HUB framework differs from prior work along three axes: (1) it models teachers as individuals rather than a distribution, enabling active selection; (2) it treats teacher selection as a sequential decision-making problem with cost-aware planning, rather than using fixed heuristics or greedy approaches; and (3) it interleaves inference and action online rather than separating them into distinct phases. We will make these distinctions explicit in the revision through a detailed comparison with each related work:
>
> - Siththaranjan et al. (2023) model teachers as a distribution rather than individuals. The HUB models them as individuals instead, which allows solution algorithms to actively select which to query.
>
> - Pitis et al. (2024) and Poddar et al. (2024) only attempt to passively identify and personalize to a given teacher (or "context," in the case of Pitis et al.). Shirali et al. (2025) assume the teacher distribution is already chosen. We instead develop solution algorithms (ATS) to actively select amongst a set of teachers, and show how this improves performance.
>
> - Siththaranjan et al. (2023), Pitis et al. (2024), Poddar et al. (2024), and Barnett et al. (2023) perform reward learning in a distinct phase. The HUB instead flexibly interleaves inference and action as a sequential decision-making process, allowing solution algorithms to only ask for feedback when necessary. Shah et al. (2020) highlights the benefits of this approach to reward learning.
>
> - Additionally, Shirali et al. (2025), Pitis et al. (2024), and Poddar et al. (2024) model only differences in preferences, whereas HUB models differences in expertise.
>
> The most closely related works are Daniels-Koch & Freedman (2022) and Barnett et al. (2023). Both model teacher selection amongst teachers with variation in expertise. However, Daniels-Koch & Freedman use a simple heuristic for teacher selection (maximizing rationality) rather than weighing expected informativeness against cost as ATS does. Section 5 of our paper discusses the limitations of this heuristic. Barnett et al. empirically demonstrate that informativeness-based selection outperforms rationality-maximizing selection, but they investigate only a greedy solution for reward inference, and do not consider query costs or impact on downstream performance as ATS does.
>
> _[continued in a second comment]_

---

> ### Author Response · Authors · 2026-02-22
> **Response to qXrq (part 2)**
>
> _[continued]_
>
> ## Change 2: Scalability
>
> We agree that discussing scalability is important. As the reviewer notes, function approximation and approximate belief updates are the key technical challenges for scaling the HUB framework beyond the current proof-of-concept domains. We will expand the Limitations and Future Work section (Section 7) to address several dimensions:
>
> - **Belief representation.** The current HUB uses exact Bayesian belief updates over discretized state spaces. For larger problems, approximate methods such as particle filtering (which is already used internally by POMCPOW) or variational inference could maintain tractable beliefs. Future improvements in large-scale POMDP solvers could be applied here.
>
> - **Planning scalability.** ATS planning time scales with the number of POMCPOW simulations, the action space size ($|\mathcal{C}| + |\beta|$), and the state space. For larger action and state spaces, techniques such as progressive widening (already part of POMCPOW) or action abstraction can help maintain tractability.
>
> - **Function approximation for reward models.** In practical RLHF settings, the reward function is typically parameterized by a neural network rather than a discrete utility table. Extending the HUB framework to this setting would require replacing exact belief updates with approximate posterior inference over network parameters (e.g., using Laplace approximation or ensembles). While this is a significant engineering challenge, the conceptual framework of modeling teacher heterogeneity and planning over teacher queries to balance informativeness against cost transfers directly.
>
> - **The framework as a modeling contribution.** We wish to emphasize that even without full POMDP planning, the HUB formalism provides value as a modeling tool. By making explicit the cost-accuracy tradeoffs between teachers, it offers a principled way to think about teacher selection that can inform simpler heuristic strategies at any scale. For example, a practitioner working with a large-scale RLHF system could use the HUB framework to reason about when to query expensive domain experts vs. cheaper crowdworkers, even if they implement the resulting policy with a simple rule rather than full POMCPOW planning.

---

> ### Author Response · Authors · 2026-03-21
> **Submission revised**
>
> Thank you again for the feedback. We've implemented these changes and uploaded a revised version of the paper. We made these improvements in response to your feedback:
>
> - Section 3.1: Clarify the distinction from related work.
> - Section 7: Expand to discuss scalability.
>
> Please let us know if you have any remaining concerns or questions.

---

### Review · Reviewer_szSi · 2026-02-12

**Summary Of Contributions:**

This paper proposes the Hidden Utility Bandit (HUB) framework to study reward learning from multiple heterogeneous teachers. In contrast to standard reward learning methods that assume a single Boltzmann-rational teacher, the proposed framework models teacher-specific rationality parameters and query costs. The authors formulate the problem as a POMDP and introduce Active Teacher Selection (ATS), which uses Monte Carlo tree search (POMCPOW) to decide both when and which teacher to query. Experiments on a recommendation task and a vaccine testing case study show that active teacher selection can improve performance compared to naïve and random strategies.

**Audience:**

Yes

**Audience Explanation:**

The problem of learning from heterogeneous teachers is important for reward modeling and RLHF. Many researchers working on interactive learning, preference learning, and human-in-the-loop systems may find the proposed framework useful. The idea of actively selecting teachers based on cost and rationality is practically relevant and timely.

**Broader Impact Concerns:**

The paper studies learning from heterogeneous teachers. While there is no immediate ethical issue in the technical formulation, modeling teacher rationality and cost may have implications in real-world human feedback systems. It would be helpful to briefly discuss potential bias or fairness concerns when aggregating heterogeneous feedback.

**Claims And Evidence:**

Yes

**Claims Explanation:**

The empirical results support the main claim that actively selecting teachers improves discounted return in the studied domains. The experimental design is generally clear, and comparisons with naïve and random baselines are appropriate.

However, the theoretical claims related to query complexity rely on the parameter $p$, defined as the worst-case teacher accuracy. Under the Boltzmann model with $\beta \geq 0$, the minimum possible accuracy is $p=1/2$, corresponding to a random teacher. The paper does not explicitly state assumptions that guarantee $p>1/2$, nor does it restrict the query profile to avoid arbitrarily small utility gaps. Without such assumptions, the theoretical bounds may not be meaningful in the worst case.

In addition, although the problem is framed as a bandit extension, the paper does not provide regret guarantees or analysis of how ATS compares to an optimal policy. This limits the theoretical contribution.

**Requested Changes:**

Below I separate critical revisions from suggestions that would strengthen the paper.

1. __Clarify identifiability assumptions in the theoretical analysis.__
The paper should explicitly state conditions ensuring $p>1/2$, such as lower bounds on teacher rationality or utility gaps. Without such assumptions, Theorems 4.3 and 4.4 may not provide meaningful guarantees.

2. __Clarify the theoretical status of ATS.__
It should be clearly stated whether ATS has any approximation guarantees relative to a Bayes-optimal policy, or whether it should be understood as a heuristic planning approach.

3. Provide discussion (even informal) comparing HUB to classical bandits and explaining why regret guarantees are difficult in this setting.

4. Include comparison with simpler heuristics (e.g., value-of-information–based strategies) to better highlight the benefit of full POMDP planning.

5. Provide more discussion on computational complexity and scalability of ATS.

---

> ### Author Response · Authors · 2026-02-22
> **Response to szSi (part 1)**
>
> We thank the reviewer for their careful and constructive review. We are glad the reviewer found the claims supported by evidence and the problem of interest to the TMLR audience, and we appreciate the suggestions to strengthen the paper. Below, we address each requested change in order.
>
> ---
>
> ## Change 1: Identifiability assumptions
>
> We thank the reviewer for identifying this gap in our presentation. The reviewer is correct that under the Boltzmann model, $p$ approaches $0.5$ as either $\beta \to 0$ or the utility gap $\Delta_{ij} \to 0$, which would make the bounds in Theorems 4.3 and 4.4 vacuous.
>
> We will add explicit assumptions to Theorems 4.3 and 4.4 to clarify when the bounds are informative. The value $p$ depends on the minimum teacher rationality and utility gap among queried pairs.
>
> 1. **Teacher rationality.** We will add an assumption requiring that the teachers have rationality $\beta > 0$, which is already implicit in any setting where querying teachers is useful (as a fully random teacher provides no information).
>
> 2. **Utility gaps.** We will add an assumption requiring that $i,j$ be ordered such that $\mathcal{U}(i) \geq \mathcal{U}(j)$, ensuring that $\Delta_{ij} \geq 0$. The query profile $Q$ determines which item pairs are compared, and in practice can be designed to avoid comparing items with near-identical utilities, ensuring that $\Delta_{ij} > 0$.
>
> ---
>
> ## Change 2: Theoretical status
>
> ATS should be understood as a principled planning approach. It uses an asymptotically optimal online solver, and we validate its finite-time performance empirically. We will add a paragraph to Section 5.1 clarifying the following:
>
> 1. **Asymptotic optimality (inherited from POMCPOW).** ATS uses POMCPOW to solve the HUB-POMDP online. POMCPOW inherits the convergence properties of POMCP (Silver & Veness, 2010): as the number of Monte Carlo simulations per decision step increases, the selected action converges to the Bayes-optimal action for the current belief state. In this sense, ATS approximates the optimal teacher selection and arm-pulling policy for any HUB instance, given sufficient computation per step.
>
> 2. **Finite-time performance.** Establishing finite-time approximation bounds would require characterizing the HUB-POMDP's specific structure (e.g., Lipschitz continuity of the value function with respect to beliefs, effective branching factor of the belief tree). We will note this as an open theoretical question. We therefore validate ATS's finite-time performance empirically, demonstrating that it outperforms baselines with practical computation budgets.
>
> ---
>
> ## Change 3: Classical bandit comparison
>
> We will add a discussion of this to Section 4. Specifically:
>
> In a standard MAB, regret is defined relative to an oracle that always pulls the best arm. In the HUB, this definition is insufficient because it does not account for the value and cost of teacher queries. An agent that never queries teachers avoids costs but will never learn the utility function; an agent that queries frequently incurs costs but gains information that improves future arm selections.
>
> The natural comparator for a HUB agent is the Bayes-optimal POMDP policy, which itself queries teachers strategically. Defining regret relative to this comparator is non-trivial, as it requires jointly reasoning about the cost of information and the value of future actions. This couples the exploration and exploitation problems in a way that classical bandit regret decompositions do not handle.
>
> ---
>
> ## Change 4: VOI heuristic comparison
>
> Barnett et al. (2023), already cited in our paper, develop a greedy, value-of-information-based teacher selection algorithm. At each step, it selects the teacher $\beta^*$ whose single query would most reduce expected belief error (MSE or log loss). This myopic strategy can be understood as a single-step-lookahead approximation to ATS's multi-step planning. However, it is not directly suitable as an empirical baseline because it operates in a pure-inference setting: it only selects which teacher to query, without arm-pulling, query costs, or an exploitation objective.
>
> We will add a discussion to Section 3.1 explicitly positioning Barnett et al. (2023) as the myopic VOI comparison and articulating the key extensions that ATS and the HUB framework provide over this approach:
>
> 1. **Multi-step planning.** Barnett et al.'s algorithm is greedy, while ATS uses POMCPOW for multi-step lookahead, allowing it to plan across the episode.
>
> 2. **Balancing exploration and exploitation.** Barnett et al. only consider which teacher to query to learn the reward function. ATS jointly decides when to gather information (exploration) and when to collect reward (exploitation).
>
> 3. **Cost-aware.** Barnett et al. treat all queries as free, while ATS incorporates teacher-specific query costs into the planning objective.
>
> _[continued in another comment]_

---

> ### Author Response · Authors · 2026-02-22
> **Response to szSi (part 2)**
>
> _[continued]_
>
> ## Change 5: Scalability
>
> We agree that discussing scalability is important and will expand the Limitations and Future Work section (Section 7) to address several dimensions:
>
> - **Belief representation.** The current HUB uses exact Bayesian belief updates over discretized state spaces. For larger problems, approximate methods such as particle filtering (which is already used internally by POMCPOW) or variational inference could maintain tractable beliefs. Future improvements in large-scale POMDP solvers could be applied here.
>
> - **Planning scalability.** ATS planning time scales with the number of POMCPOW simulations, the action space size ($|\mathcal{C}| + |\beta|$), and the state space. For larger action and state spaces, techniques such as progressive widening (already part of POMCPOW) or action abstraction can help maintain tractability.
>
> - **Function approximation for reward models.** In practical RLHF settings, the reward function is typically parameterized by a neural network rather than a discrete utility table. Extending the HUB framework to this setting would require replacing exact belief updates with approximate posterior inference over network parameters (e.g., using Laplace approximation or ensembles). While this is a significant engineering challenge, the conceptual framework of modeling teacher heterogeneity and planning over teacher queries to balance informativeness against cost transfers directly.
>
> - **The framework as a modeling contribution.** We wish to emphasize that even without full POMDP planning, the HUB formalism provides value as a modeling tool. By making explicit the cost-accuracy tradeoffs between teachers, it offers a principled way to think about teacher selection that can inform simpler heuristic strategies at any scale. For example, a practitioner working with a large-scale RLHF system could use the HUB framework to reason about when to query expensive domain experts vs. cheaper crowdworkers, even if they implement the resulting policy with a simple rule rather than full POMCPOW planning.
>
> ---
>
> ## Change 6: Bias and fairness
>
> We will add a discussion of bias and fairness considerations to Section 7. The key points are:
>
> - The HUB framework assigns rationality parameters $\beta$ to teachers, which in practice could correlate with demographic factors (e.g., domain expertise may correlate with seniority, education, or cultural background). If rationality estimates are biased, the system may systematically under-query certain groups of teachers, potentially excluding important perspectives.
>
> - The cost structure could similarly encode or reinforce existing inequities (e.g., if expert teachers are more expensive because they belong to privileged groups).
>
> - Our paper focuses on the case where teachers share a common utility function and differ only in noise and cost. The value heterogeneity setting (Section 3.1) raises additional fairness questions about *whose values* are represented, which we flag as important future work and point to Conitzer et al. (2024) for further discussion of how to integrate social choice theory and reward learning.

---

> ### Author Response · Authors · 2026-03-21
> **Submission revised**
>
> Thank you again for the feedback. We've implemented these changes and uploaded a revised version of the paper. We made these improvements in response to your feedback:
>
> - Section 3.1: Explicitly discuss Barnett et al. (2023) as the myopic VOI comparison method.
> - Section 4.2: Add explicit assumptions to clarify when the bounds in Theorems 4.3 and 4.4 are informative.
> - Section 5.1: Discuss the theoretical optimality of ATS, and explain why classical bandit-style regret guarantees are nontrivial in this setting.
> - Section 7: Discuss scalability (in a "Scalability" section) and bias and fairness considerations (in a "Societal Impact" section).
>
> Please let us know if you have any remaining concerns or questions.

---

### Review · Reviewer_5XkS · 2026-03-09

**Summary Of Contributions:**

This paper proposes the Hidden Utility Bandit (HUB) framework, which models RLHF from heterogeneous teachers differ in multiple aspects like reliability and cost. Within this framework, the authors propose Active Teacher Selection (ATS), a class of algorithms that formulate the decision of when and which teacher to query as a POMDP planning problem. This algorithm enables the agent to balance maximizing reward and improving reward model. ATS adaptively decides whether to act or request feedback, removing the need for a fixed exploration schedule and allowing selective querying of higher-quality teachers. Experiments in paper recommendation and COVID-19 vaccine show that ATS more efficiently learns the underlying utility and identifies optimal actions compared with baselines.

Strengths:
1. This paper tackles an important mismatch between the RLHF formulation (single-teacher assumption) and the reality (a heterogeneous pool of teachers), and makes meaningful contribution to address this gap.
2. Extensive empirical experiments are conducted to confirm the story in this paper, suggesting that it might be useful for RLHF in practice.

Weaknesses:
1. Maybe I am missing something, but it doesn't seem clear to me that the baseline algorithm (naive HUB inference) actually represents how the problem is typically addressed in practice.
2. The assumption that teachers share the same utility function might be strong and unrealistic.

**Audience:**

Yes

**Audience Explanation:**

This paper addresses an important problem in RLHF. Many AI/ML researchers should be interested in this paper.

**Claims And Evidence:**

Yes

**Claims Explanation:**

The claims in this paper are supported by accurate, convincing and clear evidence.

**Requested Changes:**

I recommend the authors to address my concerns listed in the "weaknesses" part above, especially the first one. It would be helpful to understand whether the baseline in the paper is actually inspired by current practice. My concern is that using fixed exploration and greedy policies are known to be weak in bandit/RL settings, so the comparison might be unfair.

---

> ### Author Response · Authors · 2026-03-09
> **Response to 5XkS**
>
> We thank the reviewer for their positive and encouraging review, and address both concerns below.
>
> ---
>
> ## 1: Current practice
>
> We appreciate this question and agree that it is important to motivate our choice of baselines clearly. We will strengthen this motivation in the revision.
>
> The naive HUB inference baseline (Algorithm 1) was designed to represent the most common preference gathering strategy in current open-source RLHF: collect a fixed batch of human preference feedback, fit a reward model to that batch, and then deploy a policy optimized against the fixed reward model. This is the standard approach described by Stiennon et al., 2020 (OpenAI), Ouyang et al., 2022 (OpenAI), Bai et al., 2022 (Anthropic), and Casper et al., 2023, and to the best of our knowledge remains the dominant paradigm in deployed systems. (It is possible that OpenAI and Anthropic have changed strategies since they stopped publishing their training details, but we can't speak to this either way.)
>
> The most recent published survey that we could find was Kaufmann et al. (2024), which notes that most systems use pre-collected preference data to train a reward model. Dong et al. (2024) advocate for online iterative RLHF, but confirm that fixed offline batches are the still default in practice. We will add these new sources to the paper to contextualize this approach.
>
> The reviewer is correct that explore-then-exploit strategies are known to be suboptimal in classical bandit settings, and this is precisely our point: the standard RLHF practice of fixed-phase feedback collection followed by deployment is an explore-then-exploit strategy applied to what is fundamentally a sequential decision problem. Our results demonstrate that actively interleaving feedback collection and deployment (as ATS does) outperforms this fixed-schedule approach, suggesting that RLHF systems could benefit from more adaptive feedback strategies.
>
> ---
>
> ## 2: Shared utility function assumption
>
> We agree that the shared utility function assumption is a meaningful restriction. We made this a deliberate scope decision in order to isolate the teacher *selection* problem: if teachers differ in both expertise and values, it becomes unclear whether querying a particular teacher is informative or simply reflects a different preference, conflating the selection problem with the value aggregation problem.
>
> The paper already discusses this in the "Value Heterogeneity" paragraph of Section 3.1, where we note that modeling value disagreement among teachers is an important but orthogonal problem rooted in social choice theory, and point to Conitzer et al. (2024) for discussion of how to integrate social choice and reward learning. In the revision, we will also add an "Ethical Considerations" paragraph to Section 7 that discusses fairness concerns arising from this assumption and flags the value heterogeneity setting as important future work.
>
> We note that the shared-utility assumption is standard in the reward learning literature — Stiennon et al. (2020), Ouyang et al. (2022), and Bai et al. (2022) all assume a single ground-truth reward function despite using multiple annotators. The HUB framework makes this assumption explicit rather than implicit, and our Section 3.1 discussion of value heterogeneity acknowledges its limitations more directly than is typical in prior work.
>
> ---
>
> ## References
>
> - Dong, H., Xiong, W., Pang, B., Wang, H., Zhao, H., Zhou, Y., Jiang, N., Sahoo, D., Xiong, C., and Zhang, T. (2024). RLHF Workflow: From Reward Modeling to Online RLHF. *NeurIPS 2024*.
> - Kaufmann, T., Weng, P., Bengs, V., and Hüllermeier, E. (2024). A Survey of Reinforcement Learning from Human Feedback. *Transactions on Machine Learning Research*.

---

> ### Author Response · Authors · 2026-03-21
> **Submission revised**
>
> Thank you again for the feedback. We've implemented these changes and uploaded a revised version of the paper. In response to your feedback, we revised Section 5.2 to motivate the "naive" family of baseline algorithms by relating to current RLHF practice. Please let us know if you have any remaining concerns or questions.

---

### Comment · Action_Editor_Yq4L · 2026-03-09
**Two weeks of discussion phase**

Dear all,

I appreciate the authors for submitting this paper and reviewers for their review efforts. Now that we have three reviews, the goal of the next couple of weeks is to have sufficient information for the reviewers to have the best recommendation.

> authors

Please address the concerns addressed by Reviewer 5XkS

> Reviewer szSi and qXrq

Please raise your remaining concerns after reading authors' responses

Best,

AE

---

> ### Author Response · Authors · 2026-03-09
> **Discussion update**
>
> Thank you very much. We have now replied to reviewer 5XkS addressing their concerns.
>
> In each reply, we describe the improvements that we plan to make to the paper to address the review comments. We believe that the changes themselves will follow straightforwardly from our replies below, but will revise the paper itself and upload it next week (by March 20).

---

> ### Author Response · Authors · 2026-03-21
> **Submission revised**
>
> We've uploaded a paper revision incorporating the reviewers' suggestions. All changes and additions are in green text. Please let us know if there are any remaining concerns or questions.

---

### Decision · Action_Editor_Yq4L · 2026-04-18

**Recommendation:** Accept as is

**Audience:**

Yes

**Audience Explanation:**

Reward learning from heterogenous sources is clearly of interest of a broad ML community.

**Claims And Evidence:**

Yes

**Claims Explanation:**

This paper considers learning reward models from multiple teachers. They propose a framework called Hidden Utility Bandit (HUB) that controls when and who to query under the POMDP model. Simulation results with COVID-19 and recommendation tasks are provided.
Reviewer 5XkS's concern is about modeling of HUB and assumption of the same utility function among teachers. The authors addressed it but Reviewer 5XkS did not provide its clear response, even though they are learning acceptance. Reviewer szSi suggested five changes. Some of them are directly addressed, the others (such as regret analysis) are discussed by authors replied that they do not consider it is of the scope of current work. Reviewer qXrq requested discussion on related work and discussion on scalability and practical extensions.

In summary, authors addressed most of the concerns carefully through the revision. As a results, all reviewers are leaning positive without specific revision request. Therefore, I support the paper.